# The Dutch Comparative Scale for Assessing Volunteer Motivations among Volunteers and Non-Volunteers: An Adaptation of the Volunteer Functions Inventory

**DOI:** 10.3390/ijerph16245047

**Published:** 2019-12-11

**Authors:** Jacobien Niebuur, Aart C. Liefbroer, Nardi Steverink, Nynke Smidt

**Affiliations:** 1Department of Epidemiology, University Medical Center Groningen, University of Groningen, PO Box 30.001, 9700 RB Groningen, The Netherlands; Liefbroer@nidi.nl (A.C.L.); n.smidt@umcg.nl (N.S.); 2Netherlands Interdisciplinary Demographic Institute, PO Box 11650, 2502 AR The Hague, The Netherlands; 3Department of Sociology, Vrije Universiteit Amsterdam, De Boelelaan 1081, 1081 HV Amsterdam, The Netherlands; 4Department of Sociology, University of Groningen, Grote Rozenstraat 31, 9712 TG Groningen, The Netherlands; b.j.m.steverink@rug.nl; 5Department of Health Psychology, University Medical Center Groningen, University of Groningen, Antonius Deusinglaan 1, AV 9713 Groningen, The Netherlands

**Keywords:** volunteer motivations, Volunteer Functions Inventory, comparative scale, psychometric properties, measurement invariance, measurement instrument, older adults

## Abstract

Currently, no valid scales exist to compare volunteer motivations between volunteers and non-volunteers. We aimed to adapt the Dutch version of the Volunteer Functions Inventory (VFI) in order to make it applicable for the comparison of volunteer motivations between Dutch older volunteers and non-volunteers. The Dutch version of the VFI was included in the Lifelines ‘Daily Activities and Leisure Activities add on Study’, which was distributed among participants aged 60 to 80. Confirmatory factor analysis (CFA) models were estimated for volunteers and non-volunteers separately, and subsequently a CFA model was created based on all observations irrespective of volunteer status. Finally, group-based CFA models were estimated to assess measurement invariance. The resulting measurement instrument (6 factors, 18 items), containing both a volunteer version and a non-volunteer version, indicated an acceptable model fit for the separate and the combined CFA models (root mean square error of approximation (RMSEA) = 0.06, comparative fit index (CFI) = 0.95). Group-based models demonstrated strong invariance between the samples. The current study provides support for the validity of the Dutch Comparative Scale for Assessing Volunteer Motivations among Volunteers and Non-Volunteers, among Dutch older adults.

## 1. Introduction

Voluntary work carries many benefits for volunteering individuals, recipients of voluntary work, organizations, and societies as a whole [1]. Benefits for volunteering older adults include the improvement of physical functioning, self-rated health, and life satisfaction, and the reduction of levels of depressive symptoms [2]. Benefits in terms of both life satisfaction, as well as perceived health, seem to be even larger for older adults than for younger adults [3]. A substantial unused volunteering capacity exists among older adults [4], especially among the growing population of retired, but still active individuals [5]. Increasing productivity in later life has societal relevance, especially in the contemporary context of population aging faced by Western societies. Increasing participation rates in voluntary work among older adults could contribute to improve the sustainability of pension and healthcare systems [6].

Previous research has investigated the sociodemographic factors associated with participation in voluntary work (for an overview, see Niebuur et al. [7]). However, the decision to volunteer is not only dependent on sociodemographic factors, but also on circumstantial, personality, and especially motivational factors [8]. So, answering the question of why some people opt for volunteering while others do not requires more insight into the motivations that people have to opt for volunteering or not. Some people may be more strongly motivated to volunteer than others and the kind of motivations may differ between people. Studying volunteer motivations can help explain why individuals participate in voluntary work, even though a lot of time and effort is required to actively seek out volunteer opportunities and commit oneself to a long-term helping relationship [9]. Studying the motivations to volunteer is important to better understand both volunteer continuation and volunteer recruitment [10] and can, therefore, help to increase participation rates.

Currently, no suitable measurement instrument is available to compare motivations to volunteer between volunteers and non-volunteers. The majority of studies on volunteer motivations are based on samples of volunteers, ignoring non-volunteering individuals [11]. As a result, little is known about the motivations to volunteer among non-volunteering individuals. Previous research comparing volunteer motivations among volunteering and non-volunteering individuals concluded that non-volunteers rate volunteer motivations lower than volunteers [12]. However, it is not clear whether the measurement instrument used allows for the comparison of volunteer motivations among these groups. Moreover, studies including volunteer samples only cannot show whether the motivations of volunteers differ from those of non-volunteers [11]. It is often implicitly assumed that volunteers have stronger motivations to volunteer than non-volunteers, and that a low level of motivations to volunteer among non-volunteers is the reason for individuals deterring from volunteering [5]. Therefore, in order to demonstrate the importance of volunteer motivations, and to eventually predict actual participation in voluntary work, we need to know whether the types of motivations to volunteer differ between volunteering and non-volunteering individuals and whether motivations to volunteer are stronger among volunteers than among non-volunteers, [11]. In order to do so, a measurement instrument allowing for the comparison of volunteer motivations between volunteers and non-volunteers is needed.

The most commonly used measurement instrument for assessing volunteer motivations is the Volunteer Functions Inventory (VFI) [10]. The VFI was developed to assess volunteer motivations in volunteering individuals [9]. In the current study, we used the VFI as a starting point and adapted it to make it applicable for comparing volunteer motivations between volunteering and non-volunteering individuals. The VFI is based on the assumption that the underlying motivations for volunteering can be distinguished into six psychological functions that can be served by volunteering [9] (p.1518). These six are: (a) The Values function: The opportunities that volunteerism provides for individuals to express values that are important to the self, such as altruistic and humanitarian concerns for others; (b) The Understanding function: The opportunity for volunteers to gain and sustain knowledge, skills, and abilities; (c) The Social function: The opportunities volunteering offer to improve social relationships, and to fit in and get along with social groups deemed important, (d) The Career function: The future job opportunities volunteering may provide; (e) The Protective function: The opportunities voluntary work offer to protect oneself from negative feelings about oneself; and (f) The Enhancement function: The opportunities participation in voluntary work offer to enhance the self-esteem by concentrating on ego growth and development.

The VFI has good psychometric properties [9] and has been shown to be applicable in different volunteer settings and in samples with different demographic characteristics. It has been translated and validated in several languages, including Chinese [13], German [14], and Dutch [15]. The VFI was developed for use in samples of current volunteers and validated among a sample of people being actively involved in volunteering. Although Clary et al. [9] also validated the VFI in a sample consisting of both people with and without volunteer experiences, no comparison between these groups was made. In another study, VFI responses of a sample of volunteers were compared to those of a sample of non-volunteers [12]. This study showed that although the ranking of the different functions served by volunteering was similar between volunteers and non-volunteers, volunteers scored higher on all volunteer motivations, except for the Career function. However, this study did not investigate whether the VFI is a valid instrument for measuring volunteer motivations in the subgroup of non-volunteering individuals. The study also did not demonstrate whether the volunteer functions measure the same underlying latent construct in the volunteer and non-volunteer samples, and whether factor mean scores can be compared between both groups. Thus, it remains unclear whether the VFI can be used to compare the types and strength of volunteer motivations between volunteers and non-volunteers. The aim of the current study was to adapt the Dutch version of the VFI in order to make it applicable for comparing the motivations to volunteer between Dutch older volunteers and non-volunteers.

## 2. Materials and Methods

### 2.1. Study Design and Participants

Lifelines is a multidisciplinary prospective population-based cohort study and biobank examining in a unique three-generation design the health and health-related behaviors of 167,729 persons living in the North of the Netherlands. It employs a broad range of investigative procedures in assessing the biomedical, sociodemographic, behavioral, physical, and psychological factors which contribute to the health and disease of the general population, with a special focus on multi-morbidity and complex genetics. The study profile of Lifelines has been described by the authors of References [16,17]. Briefly, participants were recruited between 2006 and 2013. Inhabitants (aged 25 to 50 years) of the three Northern provinces of the Netherlands were invited by their general practitioners (GPs) if they met eligibility criteria. Subsequently, respondents’ family members were invited, leading to a unique three-generation design. Additionally, inhabitants of the Northern provinces of the Netherlands could also self-register via the Lifelines website. Baseline assessment (T1) consisting of physical examinations, collecting fasting blood and urine samples, interviews, and self-report questionnaires, was conducted between 2006 and 2013. Participants were followed-up every 1.5 years by additional questionnaires, and every five years by physical examinations. All adults aged 60 to 80 who participated in the fourth Lifelines wave were invited by email to complete the electronic questionnaire for the Lifelines ‘Daily Activities and Leisure Activities add on Study (Lifelines DALAS)’. The Lifelines DALAS questionnaire composed a broad range of measures related to health, quality of life, and lifestyle, as well as a broad range of questions assessing the daily activities (i.e., employment, providing informal care and voluntary work, taking care of grandchildren) and leisure activities (i.e., sports, cultural activities, traveling, social contacts) of participants. A full section of the questionnaire was devoted to participation in voluntary work, containing questions about current and former participation in voluntary work, the frequency, duration, intensity, and type of volunteering, and the motivations underlying volunteering. The Lifelines Cohort Study was approved by the medical ethical committee of the University Medical Center Groningen, the Netherlands. All participants signed an informed consent form. Lifelines is a facility that is open to all researchers. Information on the application and data access procedure is summarized on www.lifelines.nl.

### 2.2. Adaptation of the Volunteer Functions Inventory

In order to obtain a measurement instrument allowing for the comparison of volunteer motivations between volunteers and non-volunteers, the volunteer function inventory (VFI) developed by Clary et al. [9] was used as the starting point. Several steps were taken before conducting the statistical analyses for the current study. These steps are described below and are outlined in Figure 1. The VFI [9], in which each of the six motivational functions is represented by one factor, can be found in Appendix A. It consists of 30 items, each of the six factors is represented by five items. The items are introduced by the phrase, “Please indicate how important or accurate each of the 30 possible reasons for volunteering was for you in doing volunteer work.” Each item is rated on a seven-point Likert scale, where item-score 1 represents ‘not at all important/accurate’ and item-score 7 represents ‘extremely important/accurate’. First, we translated the original VFI to the Dutch language (See Niebuur et al. [15]). Second, we adapted this instrument to make it suitable for measuring volunteer motivations among non-volunteering individuals. This adaptation was needed because some items in the original instrument were worded in such a way that they were only relevant for currently volunteering individuals. For example, item 9, originally stated as “By volunteering, I feel less lonely”, was changed into “By volunteering, I would feel less lonely” for use among non-volunteers. The following items were adapted: 1, 5, 7–15, 18, 20–22, 24–28, and 30. The other items (2–4, 6, 16–17, 19, 23, and 29) did not need any adaptation for use among non-volunteers. For example, item 6, originally stated as “People I know share an interest in community service”, is equally applicable to volunteers and non-volunteers. The introduction of the instrument was also adapted for use among non-volunteers. In the Dutch translation of the original VFI, it was stated as “Below, 30 possible reasons for participation in voluntary work are listed. Could you please indicate to what extent each of the reasons is applicable to you?” To make the introduction applicable to non-volunteers, we changed it into “Below, 30 possible reasons for participation in voluntary work are listed. What would be reasons for you to participate in voluntary work?” Volunteer status was measured in Lifelines DALAS, distinguishing volunteering individuals and non-volunteering individuals. Volunteering individuals were asked to fill-out the Dutch translation of the original VFI, and non-volunteering individuals were asked to fill out the adaptation of the instrument for use among non-volunteers.

### 2.3. Statistical Analysis

We previously validated the Dutch translation of the VFI within the Lifelines DALAS volunteer sample (see Niebuur et al. [15]). The resulting scale, referred to as the Dutch 27-item VFI-V, consists of 6 factors with 27 items in total (see Appendix A). We started with the Dutch 27-item VFI-V for volunteers (6 factors, 27 items) and the corresponding 27 adapted items for use among non-volunteers (see Appendix A).

Confirmatory factor analysis (CFA) was conducted in Stata. CFA is used to test the construct validity of the measurement instrument (i.e., to test whether groups of items can be viewed as observable indicators of unobserved underlying constructs [18]). We first estimated separate CFA models for each group (volunteers and non-volunteers), followed by a combined CFA model on all observations treated as a single group. In assessing the model fit of the separate CFA models for each group, as well as of the combined CFA model, several fit indices were used. We used the root mean square error of approximation (RMSEA) to assess absolute fit (i.e., how well a hypothesized model is able to predict the observed relationships between the data), and the comparative fit index (CFI) and the Tucker-Lewis Index (TLI) to assess incremental fit (i.e., the fit of a hypothesized model compared to the fit of a baseline model). For maximum likelihood (ML) estimation, RMSEA < 0.06 and a CFI and TLI > 0.95 indicate a relatively good model–data fit [19].

If the model fit of the separate and combined CFA models is sufficient, we can move on to testing measurement invariance between the volunteer and non-volunteer sample. Measurement invariance assesses whether estimated factors measure the same underlying latent construct within each group. As we aimed at testing measurement invariance between two groups (volunteers and non-volunteers), group-based CFA was applied. Measurement invariance is assessed by comparing models based on several levels of factorial invariance [18]. The levels of factorial invariance “form a nested hierarchy primarily represented by increasing levels of cross-group equality constraints imposed on factor loading, item intercept and residual variance parameters” [18] (page 3). Subsequently, dimensional/configural invariance, metric invariance, strong factorial invariance, and strict factorial invariance models are tested. The hierarchy of tests, imposing additional constraints on the parameters for each subsequent model, provides increasing evidence of measurement invariance. First, dimensional or configural invariance assesses whether the factor structure is equal in both groups. This is the case if the same number of common factors is present in both the volunteer and the non-volunteer sample, and all items load on the intended factors in both groups. Second, metric invariance assesses whether the common factors have the same meaning across groups. This is the case if the factor loadings are equal across groups. Third, strong invariance (or scalar invariance) assesses whether the comparison of group means is meaningful. This is the case if item intercepts are equal across groups, which means that no differential additive response bias is present in the item responses. Differential additive response bias means that forces unrelated to the common factors cause the item responses to be systematically higher or lower in one group compared to the other group. Finally, the strict invariance model further restricts the strong invariance model by imposing the residual invariance constraint to the model, which implies that corresponding item residual variances are to be equal across groups. In general, residual invariance is of limited practical value and does not contribute to support group mean comparisons [18]. As our goal was to adapt the Dutch version of the VFI to obtain a measurement instrument allowing for comparison of factor mean scores between volunteers and non-volunteers, we aimed at obtaining evidence for strong factorial invariance. So, in case the measurement instrument is demonstrated to be invariant between groups in terms of dimensional/configural variance, metric invariance, and strong invariance, sample estimates between groups can be compared. When strong invariance is confirmed, differences in sample estimates between groups reflect true group differences regarding the constructs measured [18]. In assessing whether the VFI conforms to different levels of factorial invariance, we used three fit indices. We assessed the Comparative Fit Index (CFI), Gamma hat, and McDonalds Noncentrality Index (NCI). These fit indices are independent of sample size, uncorrelated with overall fit indices, and are thus robust statistics for testing measurement invariance in group-based CFA models [20]. If the changes (∆) in the fit indices when moving from one model to the next (more stringent) model are ≤ −0.01, ≤−0.001 and ≤−0.02, for ∆ CFI, ∆ Gamma hat, and ∆ McDonalds NCI, respectively, the null hypothesis of invariance should not be rejected [20]. Finally, we assessed the reliability of the resulting scales by means of Cronbach’s α. Preferably, Cronbach’s α’s should have a value above 0.70 [21].

## 3. Results

A total of *N* = 15,655 participants was invited to participate in the Lifelines DALAS study. A total of *N* = 7639 participants filled out the questionnaire (response rate of 49.0%), with volunteer status being provided by *N* = 7612 respondents (99.6%). Of these, *N* = 4208 respondents (55.3%) indicated to participate in voluntary work at the time of filling out the questionnaire, and *N* = 3404 (44.7%) indicated not to do so. Background characteristics for the full sample (*N* = 7639), as well as the volunteer (*N* = 4208) and non-volunteer (*N* = 3404) samples, are separately presented in Table 1. As Table 1 shows, volunteers were, on average, slightly older (mean age = 67.06, SD = 4.73) than non-volunteers (mean age = 66.06, SD = 4.93). Moreover, the volunteer sample was higher educated, more often male, more often retired, and less often employed or disabled.

Descriptive statistics for all items of the VFI scales for both the volunteer sample and the non-volunteer sample are provided in Table 2. Mean and SD, median, skewness, and kurtosis are also presented. Table 2 shows that for both samples, data were highly non-normally distributed for almost every item.

### 3.1. Separate and Combined CFA Models

First, separate CFA models for both the volunteer and the non-volunteer sample were estimated. In these models, we tested the model fit for the 6-factor scales including 27 items. The results of these model tests are presented in Table 3. Fit indices for the volunteer sample suggest a nearly acceptable goodness of fit (RMSEA = 0.064, CFI = 0.899, TLI = 0.886). Fit indices for the non-volunteer sample, however, did not result in an acceptable goodness of fit (RMSEA = 0.083, CFI = 0.874, TLI = 0.857).

Second, a combined CFA model was estimated. Fit indices for all observations together treated as a single group did not result in an acceptable goodness of fit either (RMSEA = 0.072, CFI = 0.895, TLI = 0.881) (see Table 3).

### 3.2. Exploring Sources of Incomparability

The insufficient fit of the CFA models in both the non-volunteer sample and in the full sample shows that substantial differences between the volunteer sample and the non-volunteer sample exist. Therefore, we explored sources of incomparability in order to eliminate items that function differently in the volunteer and non-volunteer sample. We investigated differences between the groups for each factor separately by comparing the factor loadings from EFA analyses in both groups. Items with clearly different factor loadings were removed. The pattern matrices resulting from EFA analyses are presented in Appendix A for the volunteer sample and non-volunteer sample, respectively.

The items initially included in the Understanding factor were items 12, 14, 18, 25, and 30. Investigating factor loadings from EFA in the non-volunteer samples showed that item 12 (“I can learn more about the cause for which I am working”) was problematic because of a cross-loading (0.323) on the Career factor, and item 14 (“Volunteering allows me to gain a new perspective on things”) was problematic because of a cross-loading (0.361) on a factor containing no other items. Therefore, we eliminated items 12 and 14. 

The items initially included in the Career factor were items 1, 10, 15, 21, and 28. EFA analysis for the non-volunteer sample showed that item 21 (“Volunteering will help me to succeed in my chosen profession”) was problematic because of a cross-loading (0.301) on the Protective factor and was therefore eliminated.

The Values factor initially contained items 3, 8, 16, and 19. Factor loadings from EFA were comparable between the volunteer and non-volunteer sample and no cross-loadings were detected. However, item 8 (“I am genuinely concerned about the particular group I am serving”) had relatively low factor loadings in both samples and the interpretation could be problematic, especially among non-volunteering individuals. It could be difficult for non-volunteers to imagine the group that could potentially be served by volunteering. Therefore, item 8 was eliminated, because without item 8, the comparability of the Values factor between the groups may improve.

The items initially included in the Protective factor were items 7, 9, 20, and 24. No cross-loadings were detected. However, items 7 and 9 seemed to be problematic. The factor loadings of item 7 (“No matter how bad I’ve been feeling, volunteering helps me forget about it”) and item 9 (“By volunteering I feel less lonely”) differed substantially between the groups. The loading of item 7 was rather low in the volunteer sample (0.370) and somewhat low in the non-volunteer sample (0.535) compared to the other item loadings on the Protective factor. The factor loading of item 9 was also low in the volunteer sample (0.451) and substantially lower than the factor loading in the non-volunteer sample (0.623). Therefore, items 7 and 9 were deleted.

The factor Social initially included items 2, 4, 6, 17, and 24. Item 17 (“Others with whom I am close place a high value on community service”) had a cross-loading (0.354) on the Values factor in the non-volunteer sample. The factor loading of item 4 (“People I’m close to want me to volunteer”) was low in both the volunteer sample (0.483) and the non-volunteer sample (0.433). Therefore, items 17 and 4 were eliminated.

Finally, the Enhancement factor consisted of the items 5, 13, 26, and 27. Item 26 (“Volunteering makes me feel needed”) had a cross-loading (0.479) on the Understanding factor in the non-volunteer sample and was therefore eliminated. Item 27 (“Volunteering makes me feel better about myself”) also had a cross-loading (0.341) on the Understanding factor in the non-volunteer sample and was therefore eliminated.

The resulting scales from the investigative procedure above contained 6 factors and 18 items. We consecutively estimated CFA models for each group separately (volunteers and non-volunteers), as well as a combined CFA model including all observations treated as a single group. The results of the separate CFA models for each group and the combined CFA model on all observations are presented in Table 4. As Table 4 shows, the goodness-of-fit is sufficient in both the volunteer sample (RMSEA = 0.055, CFI = 0.951, TLI = 0.938) and the non-volunteer sample (RMSEA = 0.064, CFI = 0.949, TLI = 0.934), as well as in the combined CFA model (RMSEA = 0.058, CFI = 0.954, TLI = 0.942).

### 3.3. Group-Based CFA Models

Starting from this set of 18 items, we can now assess measurement invariance between the volunteer and non-volunteer samples. Factorial invariance was assessed by estimating several nested hierarchical models reflecting different degrees of factorial invariance. The model fit summary of all group-based analyses is presented in Table 5. These results indicate strong evidence for metric invariance (∆ CFI = −0.004, ∆ Gamma hat = -0.003 and ∆ NCI = −0.013). Although the changes in Gamma hat and McDonalds NCI (∆ Gamma hat = −0.009, ∆ NCI = −0.034) are slightly above the thresholds, as proposed by Cheung and Rensvold [20], the change in CFI (∆ CFI = −0.010) does not exceed the critical value, providing evidence for strong invariance.

The resulting measurement instrument (6 factors, 18 items), containing both a volunteer version and a non-volunteer version, seems to be a valid measurement instrument for assessing and comparing volunteer motivations among volunteering and non-volunteering individuals. This measurement instrument, the ‘Dutch Comparative Scale for Assessing Volunteer Motivations among Volunteers and Non-Volunteers’, consists of two scales: The Dutch 18-item VFI-V (for use in volunteer samples) and the Dutch 18-item VFI-NV (for use in non-volunteer samples).

To examine whether reducing the set of items (from 27 to 18 items) affected the meaning of the factors compared to how they are measured in the Dutch 27-item VFI-V, we present, in Table 6, the bivariate correlations between the factors of the Dutch 27-item VFI-V (6 factors, 27 items) and the Dutch 18-item VFI-V (volunteer version) of the comparative scale (6 factors, 18 items). Spearman’s correlation coefficients are presented because of the non-normal distribution of the data. The correlation for the Protective factor is moderate (0.80) and all other correlations are good (>0.90). Furthermore, the reliability of the Dutch 18-item VFI-V and the Dutch 18-item VFI-NV were assessed by means of Cronbach’s α. In Table 7, we present Cronbach’s α’s of the Dutch 18-item VFI-V and the Dutch 18-item VFI-NV, together with the Cronbach’s α’s of the Dutch 27-item VFI-V. The results show that all factors are internally consistent (Cronbach’s α’s > 0.70). The two scales of the Dutch Comparative Scale for Assessing Volunteer Motivations among Volunteers and Non-Volunteers are presented in Appendix B (Dutch 18-item VFI-V) and Appendix C (Dutch 18-item VFI-NV).

## 4. Discussion

The current study aimed at adapting the Dutch version of the VFI in order to make it applicable for comparing the motivations to volunteer between Dutch older volunteers and non-volunteers. Our findings provide support for the Dutch Comparative Scale for Assessing Volunteer Motivations among Volunteers and Non-Volunteers, which can be used to assess and compare volunteer motivations among Dutch volunteers and non-volunteers aged 60 to 80 years. This comparative scale, consisting of the Dutch 18-item VFI-V and the Dutch 18-item VFI-NV (6 factors, 18 items), is a valid measurement instrument. We found evidence for strong invariance, which implies that differences observed in motivations between samples of volunteers and non-volunteers reflect true differences in the importance of the motivations between the groups. This allows cross-group comparison of volunteer motivations in volunteer and non-volunteer samples.

We started from the Dutch 27-item VFI-V (the Dutch validated VFI [15]) consisting of 6 factors including 27 items. A total of nine items were eliminated in order to obtain a valid and reliable comparative scale. Of the items that were eliminated, eight out of nine items were adapted to make them assessable by non-volunteering individuals. It could therefore be that the adaptation caused a difference in interpretation between the groups, resulting in elimination of the items. The need for adapting these items already reflects that these items were originally not very well-interpretable by non-volunteers. However, the fact that they had to be removed anyway suggests that this problem persisted even after the wording of the items was changed. In Appendix A, we extensively discuss potential reasons for incomparability of the assessment of items between volunteering and non-volunteering individuals. In short, the assessment of some items by non-volunteer seemed difficult, especially when the assessment of the item required non-volunteers to imagine the specific type of voluntary work one would participate in (items 8 and 12) or to imagine how volunteering would reduce negative feelings (items 7 and 9) or increase positive feelings (items 26 and 27). Especially for non-volunteering individuals who have never volunteered before, it could be very difficult to assess these types of items, as compared to non-volunteering individuals with previous volunteer experience.

Our study shows that several items of the original VFI are not interpretable for non-volunteering individuals and that these items thus cannot be used to assess volunteer motivations among non-volunteers. This result stresses the need for a scale containing items that have been demonstrated to measure motivations to volunteer equally well for both volunteering and non-volunteering individuals when aiming to compare volunteer motivations across volunteers and non-volunteers. By not only comparing factor structures between volunteers and non-volunteers, but by also assessing whether the factor loadings and item intercepts are equal across the groups, the measurement instrument obtained in the current study allows for comparing motivations to volunteer between volunteering and non-volunteering individuals. Our study emphasizes that the original VFI by Clary et al. [9], as well as the various cross-culturally validated versions of the VFI [13,14,15], should solely be used for the purpose of assessing volunteer motivations in samples consisting of volunteering individuals. It is not meaningful to use the original VFI to assess volunteer motivations in samples consisting of both volunteers and non-volunteers, or in samples consisting of non-volunteering individuals only, because differences in factor mean scores could reflect differences in the interpretation of the items between the groups rather than true differences in volunteer motivations.

The applicability of the Dutch Comparative Scale for Assessing Volunteer Motivations among Volunteers and Non-Volunteers is twofold. First, this measurement instrument can be used to compare volunteer motivations between volunteering and non-volunteering individuals, which was the goal of the current study. Second, the Dutch 18-item VFI-V and the Dutch 18-item VFI-NV can be used to assess volunteer motivations in volunteer samples (Dutch 18-item VFI-V) and in non-volunteer samples (Dutch 18-item VFI-NV) separately, as we have demonstrated that both scales are valid and reliable measurement instruments in their own right. The Dutch 18-item VFI-V is thus a good alternative for the Dutch 27-item VFI-V, for example, in case of space constraints in questionnaires.

### 4.1. Strengths and Limitations

This is the first validated scale to assess volunteer motivations among both volunteering and non-volunteering individuals, allowing for a meaningful comparison of factor mean scores between the two groups. Major strengths of the current study are the large sample sizes (*N* = 4208 (volunteer sample) and *N* = 3404 (non-volunteer sample) and the use of a random sample of the Dutch older population. However, the current study has some limitations as well. No test–retest reliability was performed. Moreover, the comparative scale was only validated among Dutch adults aged 60 and over, so no evidence is available for the validity of the scale in other populations.

### 4.2. Recommendations for Future Research

To further examine the reliability and validity of this scale, test–retest reliability could be studied by assessing the Dutch Comparative Scale for Assessing Volunteer Motivations among Volunteers and Non-Volunteers again among the samples included in the current study. Moreover, the evidence for the validity of the scale in Dutch older adults could be improved by performing a cross-validation of the instrument in other Dutch volunteer and non-volunteer samples. Besides, the scale could be validated among samples consisting of Dutch adults ˂ 60 years of age. Finally, cross-cultural validation studies of the Dutch Comparative Scale for Assessing Volunteer Motivations among Volunteers and Non-Volunteers could be conducted in other countries to make the comparative scale available for non-Dutch study populations.

In future research, the comparative scale for assessing volunteer motivations can be used to improve the knowledge on volunteer motivations among both volunteers and non-volunteers and can reveal more insight into the differences and similarities in volunteer motivations in both groups. Previous research has revealed some indications that volunteer motivations are different among volunteering and non-volunteering individuals. Finkelstein [22] has shown that volunteer motivations within a sample of volunteers change over a 12-month period. Comparison of volunteer motivations after three months of volunteer service with those after twelve months of service revealed that volunteers initially seem to donate time, especially for other-oriented motivations, whereas volunteers seem to continue their voluntary work for a longer period of time, mainly for self-oriented motivations [22]. Using the comparative scale obtained in the current study, more insight into the underlying volunteer motivations of non-volunteering individuals can be acquired. A better understanding of the motivations to volunteer among this group could improve recruitment strategies aiming to attract potential volunteers [5,23] by offering volunteer jobs to potential volunteers that more closely fit their reasons to volunteer. Moreover, strategies aiming at retention of current volunteers, as well as motivating them to become more involved, can be optimized by better aligning the volunteer jobs with the motivations to volunteer.

## 5. Conclusions

The current study provides support for the validity of the Dutch Comparative Scale for Assessing Volunteer Motivations among Volunteers and Non-Volunteers, among Dutch older adults. This comparative scale consists of the Dutch 18-item VFI-V and the Dutch 18-item VFI-NV and allows for cross-group comparison of volunteer motivations in volunteer and non-volunteer samples.

## Figures and Tables

**Figure 1 ijerph-16-05047-f001:**
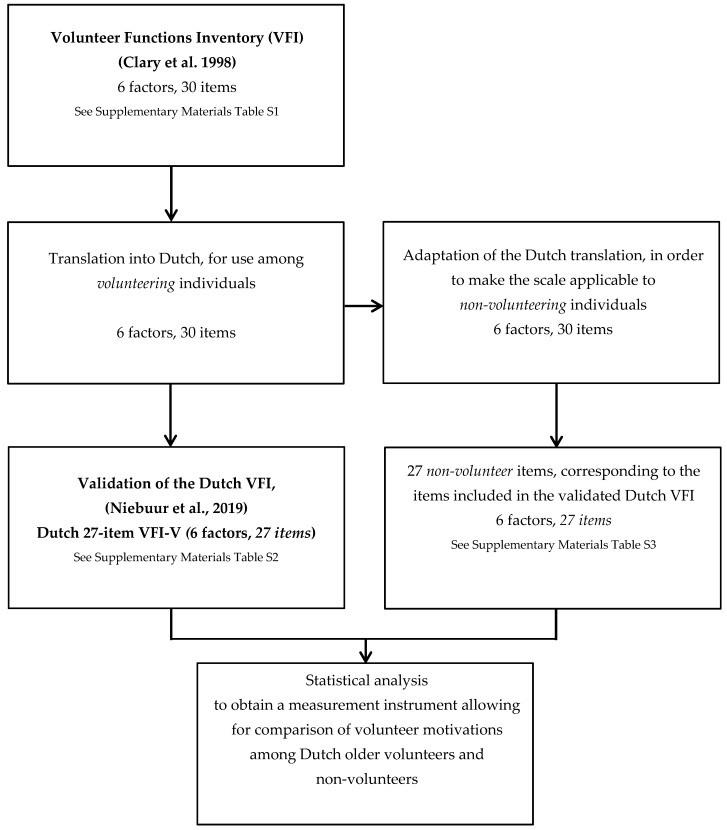
Flow diagram outlining the steps in the translation and adaptation procedure.

**Table 1 ijerph-16-05047-t001:** Descriptive statistics for the full sample, the volunteer sample, and the non-volunteer sample.

Characteristics of the Study Population	Full Sample (*N =* 7639)	Volunteer Sample (*N* = 4208)	Non-Volunteer Sample (*N* = 3404)	*p*-Value ^1^
Age, M (SD); range	66.62 (4.84); 60–80	67.06 (4.73); 60–80	66.06 (4.93); 60–79	<0.01
Gender (Female), *n* (% ^2^)	3939 (51.6%)	2123 (50.5%)	1800 (52.9%)	<0.05
Educational attainment, *n* (%)				<0.01
- Elementary - Lower secondary - Upper secondary - Tertiary	187 (2.5%) 2671 (36.1%) 2142 (28.9%) 2405 (32.5%)	81 (2.0%) 1319 (32.3%) 1173 (28.7%) 1512 (37.0%)	105 (3.2%) 1343 (40.8%) 958 (29.1%) 889 (27.0%)	
Marital status, *n* (%)				0.54
- Married/cohabiting - Relationship not cohabiting - Single/no partner	6562 (86.0%) 206 (2.7%) 867 (11.4%)	3538 (86.5%) 109 (2.6%) 459 (10.9%)	2901 (85.3%) 96 (2.8%) 405 (11.9%)	
Employment status, *n* (%) ^3^				
- Employed - Retired - Unemployed - Disabled from work	2318 (30.4%) 4859 (63.7%) 263 (3.4%) 269 (3.5%)	1,048 (24.9%) 2927 (69.7%) 158 (3.8%) 121 (2.9%)	1263 (37.2%) 1918 (56.4%) 103 (3.0%) 146 (4.3%)	<0.01 <0.01 0.08 <0.01

^1^ Obtained by conducting *χ*^2^ test; ^2^ Percentages are valid percentages (excluding missing cases). ^3^ For the employment status variables, dichotomous measures were used (employed versus unemployed, retired versus not retired, unemployed versus not unemployed, and disabled from work versus not disabled from work). The percentages in the table come from these dichotomous variables and therefore do not add up to 100.0%. Some respondents do not belong to any of these four categories and others belong to several categories (for example: A respondent can both be employed, as well as disabled from work for a certain percentage of his or her working hours).

**Table 2 ijerph-16-05047-t002:** Descriptive statistics for the Volunteer Functions Inventory (VFI) scales in the volunteer (*N* = 4208) and non-volunteer (*N* = 3404) samples.

		Volunteer Sample (*N* = 4208)				Non−Volunteer Sample (*N* = 3404)			
Subscales	Items	Mean (SD)	Median	Skewness	Kurtosis	Mean (SD)	Median	Skewness	Kurtosis
**Understanding**	12. I can learn more about the cause for which I am working	2.84 (2.00)	2	0.601 (0.04)	−1.10 (0.08)	2.08 (1.61)	1	1.314 (0.04)	0.57 (0.09)
	14. Volunteering allows me to gain a new perspective on things	3.76 (1.97)	4	−0.138 (0.04)	−1.30 (0.08)	2.27 (1.67)	1	1.042 (0.04)	−0.16 (0.09)
	18. Volunteering lets me learn things through direct, hands on experience	3.44 (1.94)	4	0.119 (0.04)	−1.31 (0.08)	2.36 (1.67)	2	1.304 (0.04)	0.77 (0.09)
	25. I can learn how to deal with a variety of people	3.91 (2.00)	4	−0.198 (0.04)	−1.28 (0.08)	2.42 (1.78)	1	0.941 (0.04)	−0.40 (0.09)
	30. I can explore my own strengths	3.23 (1.97)	3	0.245 (0.04)	−1.31 (0.08)	2.19 (1.66)	1	1.167 (0.04)	0.13 (0.09)
**Career**	1. Volunteering can help me to get my foot in the door at a place where I would like to work	1.39 (1.10)	1	3.175 (0.04)	9.98 (0.08)	1.55 (1.30)	1	2.514 (0.04)	5.44 (0.09)
	10. I can make new contacts that might help my business or career	1.60 (1.31)	1	2.363 (0.04)	4.75 (0.08)	1.57 (1.25)	1	2.404 (0.04)	5.10 (0.09)
	15. Volunteering allows me to explore different career options	1.45 (1.09)	1	2.802 (0.04)	7.67 (0.08)	1.44 (1.07)	1	2.885 (0.04)	8.58 (0.09)
	21. Volunteering will help me to succeed in my chosen profession	1.45 (1.08)	1	2.788 (0.04)	7.65 (0.08)	1.35 (0.90)	1	2.987 (0.04)	9.3 (0.09)
	28. Volunteering experience will look good on my résumé	1.57 (1.29)	1	2.495 (0.04)	5.51 (0.08)	1.56 (1.27)	1	2.464 (0.04)	5.34 (0.09)
**Values**	3. I am concerned about those less fortunate than myself	3.88 (2.17)	4	−0.118 (0.04)	−1.44 (0.08)	3.23 (1.995)	3	0.281 (0.04)	−1.26 (0.09)
	8. I am genuinely concerned about the particular group I am serving	5.38 (1.59)	6	−1.254 (0.04)	1.06 (0.08)	3.17 (2.12)	3	0.368 (0.04)	−1.34 (0.09)
	16. I feel compassion toward people in need	4.87 (1.73)	5	−0.844 (0.04)	−0.09 (0.08)	4.18 (1.85)	4	−0.329 (0.04)	−0.90 (0.09)
	19. I feel it is important to help others	5.34 (1.49)	6	−1.090 (0.04)	0.89 (0.08)	4.16 (1.86)	4	−0.297 (0.04)	−0.96 (0.09)
	22. I can do something for a cause that is important to me	4.43 (2.03)	5	−0.535 (0.04)	−1.03 (0.08)	2.56 (1.87)	2	0.827 (0.04)	−0.69 (0.09)
**Protective**	7. No matter how bad I’ve been feeling, volunteering helps me to forget about it	2.60 (1.86)	2	0.785 (0.04)	−0.73 (0.08)	1.65 (1.23)	1	2.055 (0.04)	3.72 (0.09)
	9. By volunteering I feel less lonely	2.22 (1.66)	1	1.215 (0.04)	0.32 (0.08)	1.67 (1.31)	1	2.114 (0.04)	3.82 (0.09)
	11. Doing volunteer work relieves me of some of the guilt over being more fortunate than others	1.67 (1.28)	1	2.069 (0.04)	3.55 (0.08)	1.51 (1.09)	1	2.537 (0.04)	6.49 (0.09)
	20. Volunteering helps me work through my own personal problems	1.85 (1.39)	1	1.757 (0.04)	2.34 (0.08)	1.49 (1.03)	1	2.438 (0.04)	5.95 (0.09)
	24. Volunteering is a good escape from my own troubles	1.72 (1.32)	1	2.001 (0.04)	3.33 (0.08)	1.47 (1.03)	1	2.553 (0.04)	6.39 (0.09)
**Social**	2. My friends volunteer	2.15 (1.67)	1	1.293 (0.04)	0.48 (0.08)	2.02 (1.58)	1	1.500 (0.04)	1.24 (0.09)
	4. People I’m close to want me to volunteer	1.69 (1.33)	1	2.090 (0.04)	3.65 (0.08)	1.46 (1.08)	1	2.783 (0.04)	7.77 (0.09)
	6. People I know share an interest in community service	2.65 (1.79)	2	0.685 (0.04)	−0.83 (0.08)	2.09 (1.52)	1	1.304 (0.04)	0.77 (0.09)
	17. Others with whom I am close place a high value on community service	3.57 (1.91)	4	0.013 (0.04)	−1.26 (0.08)	2.93 (1.82)	3	0.497 (0.04)	−0.94 (0.09)
	23. Volunteering is an important activity to the people I know best	2.72 (1.84)	2	0.679 (0.04)	−0.83 (0.08)	1.99 (1.50)	1	1.515 (0.04)	1.42 (0.09)
**Enhancement**	5. Volunteering makes me feel important	2.84 (1.79)	2	0.512 (0.04)	−0.10 (0.08)	1.62 (1.17)	1	2.058 (0.04)	3.78 (0.09)
	13. Volunteering increases my self−esteem	3.30 (1.93)	3	0.197 (0.04)	−1.28 (0.08)	1.82 (1.39)	1	1.706 (0.04)	2.00 (0.09)
	26. Volunteering makes me feel needed	3.81 (1.89)	4	−0.176 (0.04)	−1.20 (0.08)	2.14 (1.59)	1	1.210 (0.04)	0.30 (0.09)
	27. Volunteering makes me feel better about myself	3.39 (1.90)	4	0.108 (0.04)	−1.28 (0.08)	1.90 (1.44)	1	1.554 (0.04)	1.46 (0.09)
	29. Volunteering is a way to make new friends	3.35 (1.95)	3	0.167 (0.04)	−1.29 (0.08)	2.60 (1.82)	2	0.740 (0.04)	−0.74 (0.09)

**Table 3 ijerph-16-05047-t003:** Model fit summary confirmatory factor analysis (CFA; 6 factors, 27 items).

**Separate CFA models**	***N***	**RMSEA**	**CFI**	**TLI**
Volunteer sample	4010	0.064	0.899	0.886
Non-Volunteer sample	3115	0.083	0.874	0.857
**Combined CFA model**	***N***	**RMSEA**	**CFI**	**TLI**
Full sample	7125	0.072	0.895	0.881

**Table 4 ijerph-16-05047-t004:** Model fit summary CFA (6 factors, 18 items).

**Separate CFA Models**	***N***	**RMSEA**	**CFI**	**TLI**
Volunteer sample	4043	0.055	0.951	0.938
Non-Volunteer sample	3129	0.064	0.949	0.934
**Combined CFA model**	***N***	**RMSEA**	**CFI**	**TLI**
Full sample	7172	0.058	0.954	0.942

**Table 5 ijerph-16-05047-t005:** Group-based CFA models (6 factors, 18 items).

Group-based CFA Models.	*N*	RMSEA	CFI	TLI	Gamma Hat	NCI	Reference Model #	Δ CFI	Δ Gamma Hat	Δ NCI
1. dimensional/ configural invariance	7172	0.059	0.950	0.936	0.956	0.812				
2. metric invariance	7172	0.060	0.946	0.935	0.953	0.799	1	−0.004	−0.003	−0.013
3. strong invariance	7172	0.064	0.936	0.926	0.944	0.765	2	−0.010	−0.009	−0.034

**Table 6 ijerph-16-05047-t006:** Bivariate correlations between the Dutch 27-item VFI-V and the Dutch 18-item VFI-V (Volunteer sample).

Subscales	Spearman’s Rho
Understanding	0.94
Career	0.98
Values	0.97
Protective	0.80
Social	0.94
Enhancement	0.97

**Table 7 ijerph-16-05047-t007:** Internal consistency (Cronbach’s Alpha).

Subscales	Dutch 27-item VFI-V (Volunteers)	Dutch 18-item VFI-V (Volunteers)	Dutch 18-item VFI-NV (Non-Volunteers)
Understanding	0.83	0.78	0.84
Career	0.85	0.81	0.84
Values	0.78	0.77	0.81
Protective	0.81	0.86	0.87
Social	0.78	0.70	0.71
Enhancement	0.85	0.80	0.84

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
