# Peer review of "The Dutch Comparative Scale for Assessing Volunteer Motivations among Volunteers and Non-Volunteers: An Adaptation of the Volunteer Functions Inventory"

_ijerph, 2019, doi:10.3390/ijerph16245047_

Round 1

Reviewer 1 Report

it is a good study to examine the motivation being a volunteers and non-volunteers, the findings can be contributed to the literature. 

Introduction: please add more information about the non-volunteers, any literature about this ?

Methods: may need a bit details in approaching the participants;    

Results & Discussion : adequately addressed the research aims & objectives; and, in the discussion, please link up the results with the significance of study, and further plan/action with the study results, eg: how are you going to motivate those non-volunteers to become more involve / becoming volunteers etc....  

Reviewer 2 Report

This article reports a complex and good study. The authors have done a very good job. The results are particularly important for the Dutch volunteer sector.

All in all, the article is well structured, understandable and readable. Background of the study is solid and well described till line 52. I recommend the authors to revise the content between lines 53 to 65 in order to optimize or to shortened it. It is argued that there is currently no suitable instrument to assess the motivations between volunteers and non-volunteers. Subsequently, the objective of developing an appropriate instrument is presented. But the objective in line 54 is not as precise the subject of this study, as the Dutch population is not pointed out as the target group of this study. In addition the objectives in Line 53 represents an unnecessary duplication to the objectives shown in Line 94.

The target in line 94 does not fit to the following sentence from line 96. If an existing scale is changed, it is called an adaptation and not a development of a new scale. Even if the FVI was extensively revised, it is still an adaptation and not a new development of a scale. Even though the majority of items have been revised and new ones added, it is still an adaptation. A new development requires a completely different research design. 

Recommendation to the authors: Either you explain very precisely why it is in your view a new development of an instrument, or you change the title, correct the introduction and the discussion and report of an adaptation.

In my opinion, the clarification of an adaptation will greatly enhance the value of the article. Once again, I would like to pay tribute to the good study. All the best for your work.
